# Effect of Stepwise Exposure to High-Level Erythromycin on Anaerobic Digestion

**DOI:** 10.3390/molecules29153489

**Published:** 2024-07-25

**Authors:** Yanxiang Zhang, Chunxing Li, Xinyu Zhu, Irini Angelidaki

**Affiliations:** 1School of Environmental and Material Engineering, Yantai University, Yantai 264005, China; 15092166641@163.com; 2State Key Laboratory of Pollution Control and Resource Reuse, School of the Environment, Nanjing University, Nanjing 210023, China; wflqlcx@163.com; 3Key Laboratory of Coastal Environment and Resources of Zhejiang Province, School of Engineering, Westlake University, Hangzhou 310030, China; 4Department of Chemical and Biochemical Engineering, Technical University of Denmark, DK-2800 Kongens Lyngby, Denmark; iria@kt.dtu.dk

**Keywords:** anaerobic digestion, erythromycin, antibiotic resistance genes, high-level antibiotic, microbial community

## Abstract

High-level erythromycin (ERY) fermentation wastewater will pose serious threats to lake environments. Anaerobic digestion (AD) has advantages in treating high-level antibiotic wastewater. However, the fate of antibiotic resistance genes (ARGs) and microbial communities in AD after stepwise exposure to high-level ERY remains unclear. In this study, an AD reactor was first exposed to 0, 5, 10, 50, 100 and 200 mg/L ERY and then re-exposed to 0, 50, 200 and 500 mg/L ERY to investigate the effect of ERY on AD. The results show that AD could adapt to the presence of high-level ERY (500 mg/L) and could maintain efficient CH_4_ production after domestication with low-level ERY (50 mg/L). The AD process could achieve higher removal of ERY (>94%), regardless of the initial ERY concentration. *ErmB* and *mefA*, conferring resistance through target alteration and efflux pumps, respectively, were dominant in the AD process. The first exposure to ERY stimulated an increase in the total ARG abundance, while the AD process seemed to discourage ARG maintenance following re-exposure to ERY. ERY inhibited the process of acetoclastic methanogenesis, but strengthened the process of hydrogenotrophic methanogenesis. This work provides useful information for treating high-level ERY fermentation wastewater by the AD process.

## 1. Introduction

Erythromycin (ERY), a broad-spectrum macrolide antibiotic, is produced by the fermentation of *Streptomyces* species [1]. Since ERY is frequently used in human medicine and veterinary medicine, its demand is high. For example, the usage of ERY in the UK is high, at 48 tonnes/year [2]. However, due to incomplete extraction during the ERY production process, its residue in fermentation wastewater is high, reaching the 100–500 mg/L level [3]. ERY has been detected globally in sewage (max. conc. of 1037 μg/kg dry wt.), surface water (max. conc. of 3847 ng/L) and sediment (max. conc. of 175.38 μg/kg dry wt.) [4]. The accumulation of ERY in the environment may pose a significant risk to the ecosystem. Previous studies found that ERY could cause toxic effects on algae, such as *Microcystis flos-aquae* [5] and *Pseudokirchneriella subcapitata* [6], as well as fish, such as *Sparus aurata* L. [7] and *Oncorhynchus mykiss* [8]. ERY is included on the US EPA’s Drinking Water Contaminant Candidate List 4 (CCL-4) [9] and the EU’s Water List [10]. Therefore, it is urgent to deal with high-level ERY fermentation wastewater.

Various treatment methods have been used to remove ERY from wastewater [11,12,13]. For example, Gholamiyan et al. used magnetic-activated carbon as an adsorbent for the removal of ERY and found that 95% of ERY could be adsorbed [11]. Mohammed et al. synthesized CuFe_2_O_4_@Chitosan for photocatalytic degradation of ERY and found a high ERY degradation efficiency (90%) [12]. Chu et al. investigated the removal of ERY by ionizing radiation and found that the ERY removal rate could reach 87% [13]. However, the above treatment methods are expensive. In addition, the high chemical oxygen demand (COD) and total suspended solids (TSSs) in ERY fermentation wastewater limited their application in ERY wastewater treatment. Anaerobic digestion (AD) is considered an economically feasible method to deal with recalcitrant pollutants, owing to its low cost and high energy efficiency [14,15]. Spielmeyer et al. [16] investigated the removal of tetracyclines and sulfonamides by anaerobic fermentation and found that their elimination rates ranged from 14% to 89%. It has been reported that the AD process could achieve 100% elimination of tetracycline [17].

Nowadays, there are, however, only limited studies investigating the treatment of high-level ERY fermentation wastewater by the AD process. Zeynep Cetecioglu et al. [18] found that methane generation was almost completely inhibited when the ERY dosage was above 500 mg/L. They also found that low-level ERY (1–100 mg/L) did not affect the removal of volatile fatty acids, and high-level ERY (250–1000 mg/L) impaired the utilization of propionate and reduced the removal of butyrate [19]. The above-mentioned research mainly focused on the transformation of organic matter. Actually, during the AD process, high-level ERY may hinder the growth of bacteria by binding to the 50S ribosomal subunits, further affecting the microbial community structure [20]. Moreover, the presence of high-level ERY may induce or affect the amplification of antibiotic resistance genes (ARGs) (four types, i.e., efflux genes, methylase genes, phosphorylase genes and esterase genes) [21]. Wang et al. reported that high-level ERY caused an increase in the relative abundance of *Sedimentibacter* and *Methanosarcina* and accelerated methylase gene (*erm A/T*) amplification [22]. The relative abundance of *Syntrophomonas*, *Syntrophorhabdus* and *Methanosaeta* increased under the pressure of high-level clarithromycin (an ERY derivative) [23]. Methylase genes (*ermF* and *ermB*) were enriched with the addition of high-level lincomycin (macrolide–lincosamide–streptogramin (MLS) antibiotic) [24]. However, the mentioned study only evaluated the influence of the first exposure to high-level ERY on the AD process. Up to now, the effect of stepwise exposure to high-level ERY on ARGs and microbial communities of the AD system has been little reported.

In order to investigate the influence of stepwise exposure to high-level ERY on anaerobic fermentation, an AD reactor was first exposed to 0, 5, 10, 50, 100 and 200 mg/L ERY, and then re-exposed to 0, 50, 200 and 500 mg/L ERY. The main objectives of this study were to (1) explore methane production during stepwise exposure to high-level ERY; (2) evaluate the removal efficiency of ERY in the AD system; (3) investigate the fate of key ARGs’ abundance and discuss the resistant mechanism; and (4) determine the diversity and structure of the microbial communities. Our study will provide a theoretical basis for the treatment of high-level ERY fermentation wastewater by the AD process.

## 2. Results and Discussion

### 2.1. Methane Generation Performance in the Existence of ERY

Methane production is usually used as an indicator of AD performance. Figure 1 displays the effects of ERY on methane production during AD. As shown in Figure 1a, in the presence of 5 and 10 mg/L ERY, CH_4_ production was higher than that of the glucose control group during the whole experiment, indicating that low levels of ERY would promote CH_4_ production. Low concentrations of ERY may have limited the antibacterial spectra of the bacteria community [25]. Mustapha et al. [26] found that macrolide antibiotics (ERY at 15 mg/L, clarithromycin at 15 mg/L and roxithromycin at 15 mg/L) promoted CH_4_ production. However, when the concentration of ERY increased further (50, 100 and 200 mg/L), decreases in CH_4_ production were observed during the experiment. Therefore, it could be speculated that high levels of ERY could inhibit the activity of CH_4_ production. In addition, the ultimate CH_4_ production remained at similar levels, indicating that the inhibition effect of ERY on AD progressively recovered with extended exposure. The recovery of CH_4_ production might be due to the domestication of certain bacteria or the removal of ERY. 

According to Figure 1b, 50–50 and 200–200 showed no apparent inhibition effects on CH_4_ production. This might be because bacteria had already adapted to the same concentration of ERY during phase Ⅰ inoculation. In addition, the inhibition effect of 0–500 on CH_4_ production was similar to that of high-level ERY in phase Ⅰ. Interestingly, CH_4_ production remained relatively efficient in 50–500, while it was negatively affected by 200–500. A low concentration of the antibiotic is more likely to foster microbial resistance [27]. Fan et al. [28] studied the causal correlation of ERY with resistance proliferation. The result suggested that the presence of ERY at low concentrations could increase the establishment of antibiotic resistance. Thus, compared with 200, bacteria might find it easier to develop resistance under 50 in phase Ⅰ and could quickly adapt to 500 in phase II. 

### 2.2. The Removal of ERY during AD

Figure 2 illustrates the variation in ERY concentration during AD. After the anaerobic reaction of phase Ⅰ, the removal efficiency of ERY in 50 and 200 reached 95.9% and 99.8%, respectively. The results imply that ERY could be effectively removed, even though ERY was difficult to biodegrade under anaerobic conditions. Feng et al. [29] examined the removal of ERY during anaerobic digestion of pig manure. In their study, the removal efficiency of ERY reached 99.9% at day 40, which indicated that this antibiotic could be effectively degraded in the AD system. In addition, the above results show that the increase in ERY concentration led to the improvement in removal efficiency. One possible explanation is that the antibiotic as a carbon source was more likely utilized by microorganisms at higher concentrations. Yang et al. [30] explored the effect of antibiotic concentration on the removal of antibiotics and found that the removal efficiency of antibiotics (sulfamethoxazole, chlorotetracycline and penicillin) improved with increasing antibiotic concentration in a sequencing batch reactor. This might be the reason that the inhibition effect of ERY gradually weakened and could recover when prolonging the exposure time. 

After the anaerobic reaction of phase II, the removal efficiency of ERY was still high. The removal efficiency of ERY was 94.5% in 0–500, which was not consistent with the concept that the higher the antibiotic concentration, the better the removal efficiency. This was likely due to the microbial acclimation in phase II or the inhibition of microorganism activity under a higher concentration of ERY. In addition, the removal efficiency of ERY in 200–500 (96.4%) was higher than that in 50–500 (94.5%), but CH_4_ production in 200–500 was lower than that in 50–500. It could be speculated that the efficient CH_4_ production in 50–500 might not be caused by ERY removal, but by bacteria domestication under ERY pressure during phase I.

### 2.3. Variations in ARG Types and Relative Abundance

Ten ERY resistance genes, including two phosphorylase genes (*mphA-01*, *mphB*), two esterase genes (*ereA*, *ereB*), four methylase genes (*ermA*, *ermB*, *ermC*, *ermX*) and two efflux pump genes (*mefA*, *mefE*), were detected. As shown in Figure 3, *ermB* and *mefA* were abundant in 0. The former confers resistance to ERY by target alteration due to methylation, while the latter pumps ERY into the extracellular environment [31]. The results indicate that target alteration and efflux pumps were the dominant resistance mechanisms against ERY in the original sludge. The total ARG abundance was enhanced under ERY loading conditions in phase I, which might be owing to the increase in ERY selection pressure on the microbial communities. Meanwhile, the addition of ERY substantially increased the relative abundance of *ermB* and *mefA*, suggesting that ERY exposure improved the influence of methylation and efflux pumps. However, phosphorylase genes and esterase genes displayed insignificant changes, as well as the remaining efflux pump genes and methylase genes. The phosphorylase genes and esterase genes could inactivate ERY by encoding ERY phosphotransferase and esterase, respectively [19]. The results demonstrate that antibiotic inactivation played a minor role in ERY resistance.

It is clear that *ermB* and *mefA* were still dominant in phase II. The total ARG abundance in 0–0 was much higher than that in 0, which might be caused by the microbial acclimation. In addition, 0–500 presented the highest enrichment of total ARGs. Interestingly, the total ARG abundance in 50–50 and 200–200 in phase II decreased compared with those in 50 and 200 in phase I, respectively. This might be due to the fact that the first exposure of the antibiotic was more likely to stimulate an increase in the total ARG abundance. Moreover, in comparison with 200–200 and 200–500, the total ARG abundance was much lower in 50–50 and 50–500. The ARGs expressed in response to ERY could be adapted and reserved in 50. It was likely that these ARGs were less sensitive to ERY pressure in 50–500. 

### 2.4. Variations in Microbial Community Diversity and Structure

#### 2.4.1. Microbial Community Diversity

The richness and diversity of the microbial communities were estimated using the alpha diversity indices. As shown in Figure 4a, the Chao1, Observe, Shannon and Simpson indices of 50 were smaller than those of 0, suggesting that the microbial richness and diversity decreased. A possible explanation is the adaptation of certain bacteria and the inhibition of other bacteria under 50. Unlike 50, the microbial richness and diversity increased in 200, which could be explained by the removal of ERY. A previous study showed that bacteria diversity was positively related to the biotransformation rate of micro-pollutants [32]. In addition, the above four indices of 0–0 decreased compared with those of 0. The addition of ERY reduced the microbial richness and diversity in phase II, among which 0–500 presented the lowest levels, indicating that microorganism activity was inhibited under a higher concentration of ERY. On the other hand, microbial richness and diversity indices of 50–500 were higher than those of 50–50, 200–200 and 200–500, which demonstrated the adaptation of certain bacteria during phase I. The similarity in the microbial communities was revealed by PCoA (Figure 4b). PCoA1 and PCoA2 together explained 56.47% of the total variation. As we can see, 0–0 was deviating considerably from 0, which was caused by the microbial acclimation. The microbial communities of 0 were similar to those of 50, but different from those of 200. In addition, PCoA did not reveal any difference between 0–0 and 50–500, but revealed a difference between 0–0 and 0–500. These results were consistent with the changes in CH_4_ production.

#### 2.4.2. Microbial Community Structure 

The microbial community composition was analyzed at the phylum and genus levels (Figure 5). The phyla in all of the samples contained Bacteroidota (17.6–55.6%), Firmicutes (17.3–47.4%), Synergistota (7.9–28.2%), Caldatribacteriota (0.8–7.2%), Cloacimonadota, Halobacterota, Proteobacteria, Thermotogota and WS1. Bacteroidota could degrade glucose, glycerol and amino acids to produce volatile fatty acids in the anaerobic digestion process [33]. Moreover, Bacteroidota were correlated with the hydrolysis of macromolecular organic compounds [33]. Firmicutes played a major part in hydrolysis and acidogenesis [13]. Additionally, Firmicutes were involved in acetogenesis/syntrophic acetate oxidation in the AD system [13]. Synergistota contained a large number of organic matter-degrading bacteria, contributing to the production of methane [34]. Bacteroidota, Firmicutes and Synergistota were often found in anaerobic digesters owing to their ability to decompose a range of organic compounds [35]. The phylum Halobacterota was the dominant archaea species in all of the samples, which is known as a group of methanogens [36]. Most methanogens in the phylum Halobacterota were reported to use methanol, acetate and hydrogen as electron donors. As shown in Figure 5a, Bacteroidota increased in 50 but decreased in 200, which might be due to the fact that the hydrolytic process associated with Bacteroidota was promoted at low-level ERY but inhibited at high-level ERY. The addition of ERY and the microbial acclimation both affected the composition of phyla, which is in agreement with the alpha diversity results (Figure 4a). 

At the genus level, *Proteiniphilum* and *Acetomicrobium*, typical hydrolytic/fermentative bacteria, were dominant in all of the samples. In phase I, the relative abundance of *Lentimicrobium*, *WS1* and *Methanosarcina* displayed a decreasing trend with the increase in ERY concentration, which demonstrated the inhibition effects of ERY on these bacteria. *Lentimicrobium* is a strictly anaerobic bacterium that can degrade glucose to acetate, malate, propionate, formate and hydrogen [37]. *Methanosarcina* is one kind of acetoclastic methanogen and can metabolize acetate to produce CH_4_ [38]. Its decrease indicated that ERY inhibited the acetoclastic methanogenesis. The relative abundance of *DTU014*, *Aminobacterium* and *Alkaliphilus* maintained an increasing trend under ERY pressure, indicating that these bacteria could resist the toxicity of ERY. *Aminobacterium*, one kind of amino-acid-degrading bacterium, is closely related to hydrogenotrophic methanogens [39]. *Alkaliphilus* is capable of producing glycoside hydrolase under anaerobic fermentation [40], which may contribute to the removal of ERY. This is because ERY, composed of the hexose sugar cladinose, is a candidate target of glycosylase enzymes [41]. In addition, *DTU014* and *Alkaliphilus* have shown potential as syntrophic acetate-oxidizing bacteria [42], which could couple to perform hydrogenotrophic methanogenesis. The increased relative abundance of the above genera might be the reason for the recovery of CH_4_ production under 200. 

In phase II, *Fermentimonas* and *Acholeplasma*, typical hydrolytic/acetogenic bacteria, were enriched when glucose changed into avicel. The relative abundance of *MgMjR-022*, *Ruminofilibacter*, *Syntrophomonas* and *Proteiniborus* in 0–500 was higher than that in 0–0, which demonstrated that they could endure ERY biotoxicity. On the contrary, the relative abundance of *Fermentimonas* and *Acholeplasma* in 0–500 was lower than that in 0–0. *Fermentimonas* can use carbohydrates and complex proteinaceous to produce acetic acid, hydrogen and carbon dioxide [43]. *Acholeplasma* produces fatty acids that can be used for methanogenesis [44]. The reduced relative abundance of these genera suggests that hydrolysis and acidogenesis were affected by ERY, in turn leading to the inhibition of CH_4_ production. It should be noted that the relative abundance of *MgMjR-022*, *Ruminofilibacter*, *Fermentimonas*, *Acholeplasma* and *Aminobacterium* was still quite high in 50–50, 50–500 and 200–200. Hence, the high relative abundance of hydrolytic and acetogenic bacteria might contribute to the production of CH_4_. Furthermore, *Methanoculleus* showed the highest relative abundance in 50–500. *Methanoculleus* is responsible for hydrogenotrophic methanogenesis and is correlated with syntrophic acetate oxidation [45]. The result suggests that hydrogenotrophic methanogens were facilitated under 50–500.

Overall, the effect of stepwise exposure to high-level ERY on AD resulted in the enrichment of *Aminobacterium*, *MgMjR-022* and *Ruminofilibacter*. Those enriched bacteria might be associated with the changes in ARGs to a certain extent. In addition, based on the above analysis, it was inferred that the hydrogenotrophic methanogenesis pathway might be less disturbed than the acetoclastic methanogenesis pathway under 50 in phase I. A previous study found that hydrogenotrophic methanogens were less sensitive to 50 mg/L of ciprofloxacin compared with acetoclastic methanogens [33]. Thus, hydrogenotrophic methanogens in 50 could quickly adapt to the high-level ERY in phase II and played an important role in CH_4_ production.

### 2.5. Assessment of the Influence of Microbial Community on ARGs

The microbial community is believed to influence the variation in ARGs [46]. Network analysis is an essential method to reveal the correlation between ARGs and the microbial community [47]. In this study, the relative abundance of *mefA* was high, but the correlation between *mefA* and the microbial community was not found through network analysis, which might be caused by the small sample size. However, it has been reported that *Fermentimonas* was a potential host of *mefA* [48]. Thus, the increase in the abundance of *mefA* in 0–0 might be related to the enrichment of *Fermentimonas*. In addition, the increased abundance of *ermB* could be explained by the proliferation of the potential hosts, such as *Alkaliphilus* and *Ruminofilibacter,* according to previous reports [49]. Moreover, *Acholeplasma* was identified as a co-host of *ermB* and *mefA* which is saprotrophic and pathogenic [49]. *Alkaliphilus* and *Acholeplasma* belong to Firmicutes, while *Fermentimonas* and *Ruminofilibacter* belong to Bacteroidota. Those two phyla were reported to be responsible for carrying and transferring ARGs in a variety of environments [50]. In this study, *mefA* and erm B, as well as most of the above-mentioned hosts, were not enriched in 50–500, suggesting that bacteria were better adapted to the presence of the high-level antibiotic in phase II after domestication with the low-level antibiotic in phase I.

## 3. Materials and Methods

### 3.1. Experiment Setup

The inoculum used in the experiment was obtained from an anaerobic reactor treating sewage sludge at 37 °C. After storage in the incubator for 1 month, its methane potential was removed. The total solid (TS) of the inoculum was 17.8 g/L; the volatile solid (VS) was 10.6 g/L; and the pH was 8.40. Erythromycin (C_37_H_67_NO_13_) was purchased from Aladdin Chemistry Co., Ltd. (Shanghai, China).

Batch tests were carried out in 118 mL serum bottles. Two phases were included in the experiment. The procedure of phase I was as follows: 10 mL inoculum was added into the serum bottle; glucose was used as a carbon source to achieve the required COD concentration (5000 mg/L); 20 mL ERY was injected into the bottle to obtain a range of 0, 5, 10, 50, 100 and 200 mg/L. A blank test was performed with only inoculum. Thereafter, those bottles were purged with nitrogen to exclude oxygen and then closed using butyl rubber stoppers and aluminum crimps. The bottles were named blank, 0, 5, 10, 50, 100 and 200. Then, the prepared bottles were placed in an incubator at 37 °C. The experiment was terminated when the gas production was insignificant. All of the experiments were carried out in triplicate. The digestate from phase I (0, 50 and 200) was used as inoculum in phase II. The procedure of phase II was as follows: Avicel was used as a carbon source to achieve the required COD concentration (5000 mg/L). Then, 10 mL products of 0, 50 and 200 were spiked with 20 mL ERY in new bottles to obtain a range of 0 and 500 mg/L (named 0–0 and 0–500), 50 and 500 mg/L (named 50–50 and 50–500) as well as 200 and 500 mg/L (named 200–200 and 200–500). The remaining steps were the same as in phase I. In addition, the concentration of ERY was determined at the end of the experiment. Cs and Ce represent the initial and final concentrations of ERY, respectively. As for ARGs and microbial analysis, the products of 0, 50, 200, 0–0, 0–500, 50–50, 50–500, 200–200 and 200–500 were taken from the reactor. 

### 3.2. Microbial Analysis 

Firstly, the sample was washed using phenol to improve DNA purification. Then, DNA was extracted with DNeasy PowerSoil^®^ (QIAGEN GmbH, Hilden, Germany). DNA quality was evaluated with NanoDrop (Thermo Scientific, Waltham, MA, USA). After that, 16S rRNA genes were amplified using the primers 515F/806R. Sequencing was conducted using the Illumina high-throughput sequencing technology [51]. 

### 3.3. ARG Analysis

The abundance of the ERY resistance gene was examined through RT-qPCR using the StepOnePlus™ Real-Time PCR system (Wcgene Biotechnology, Shanghai, China). The target genes included *ereA*, *ereB*, *mphA-01*, *mphB*, *ermA*, *ermB*, *ermC*, *ermX*, *mefA* and *mefE*. The ERY resistance gene primers are shown in the Supplementary Data. The detailed composition of the PCR mixture and the PCR procedures were processed according to a previous study [52]. The efficiency of amplification should be within 1.8–2.2. The relative abundance of ARGs was normalized to 16S rRNA genes, which was measured according to a comparative CT method, 2^−ΔCT^, where ΔCT = (CTARG − CT16S rRNA gene).

### 3.4. Analytical Methods

TS and VS were analyzed according to [53]. The pH was measured by a pH meter (FiveEasy Plus Benchtop FP20, Mettler Toledo, Greifensee, Switzerland). The content of CH_4_ was quantified by using a gas chromatograph (Trace 1310 GC-TCD, Thermo Fisher, Lillerød, Denmark) equipped with a thermal conductivity detector (TCD) and Thermo (P/N 26004–6030, Thermo Scientific, Waltham, MA, USA) Column (30 m length, 0.320 mm inner diameter and film thickness 10 µm) with helium as the carrier gas. The concentration of ERY was determined using a high-performance liquid chromatography (Agilent 1290 Infinity, Santa Clara, CA, USA, HPLC) system coupled with a tandem mass spectrometer (Agilent 6470 series, USA, MS/MS). The system consisted of electrospray ionization (ESI) and a C18 column (2.1 × 50 mm, 1.7 μm). The positive ion mode (ESI^+^) was selected. ERY was detected by the prominent fragment ion at *m*/*z* 158. Mobile phase A was MeOH and mobile phase B was water plus 0.1% formic acid (at a flow rate of 0.2 mL/min for 8 min). The elution gradient mode was as follows: 45% A (0–2.5 min), 25% A (2.5–4 min), 10% A (4–7 min) and 25% A (7–8 min). Principal coordinate analysis (PCoA), a heatmap and a chord diagram were conducted using R software (4.2.0). OriginPro 2021 was used to analyze other figures.

## 4. Conclusions

This study investigated the influence of stepwise exposure to high-level ERY on anaerobic digestion, and the main conclusions are as follows: (1) After domestication with low-level ERY, AD could adapt to the presence of high-level ERY and could maintain efficient CH_4_ production. (2) The AD process could achieve higher removal of ERY. (3) The first exposure to ERY stimulated an increase in the total ARG abundance, while the AD process seemed to discourage ARG maintenance following re-exposure to ERY. (4) Under the influence of ERY, the hydrogenotrophic methanogenesis pathway replaced the acetoclastic methanogenesis pathway for CH_4_ production. There are still some limitations in this study: the effect of the transformation products of ERY on the fate of ARGs and microorganisms needs further research, and the application of AD for the treatment of actual antibiotic fermentation wastewater should be clarified. 

## Figures and Tables

**Figure 1 molecules-29-03489-f001:**
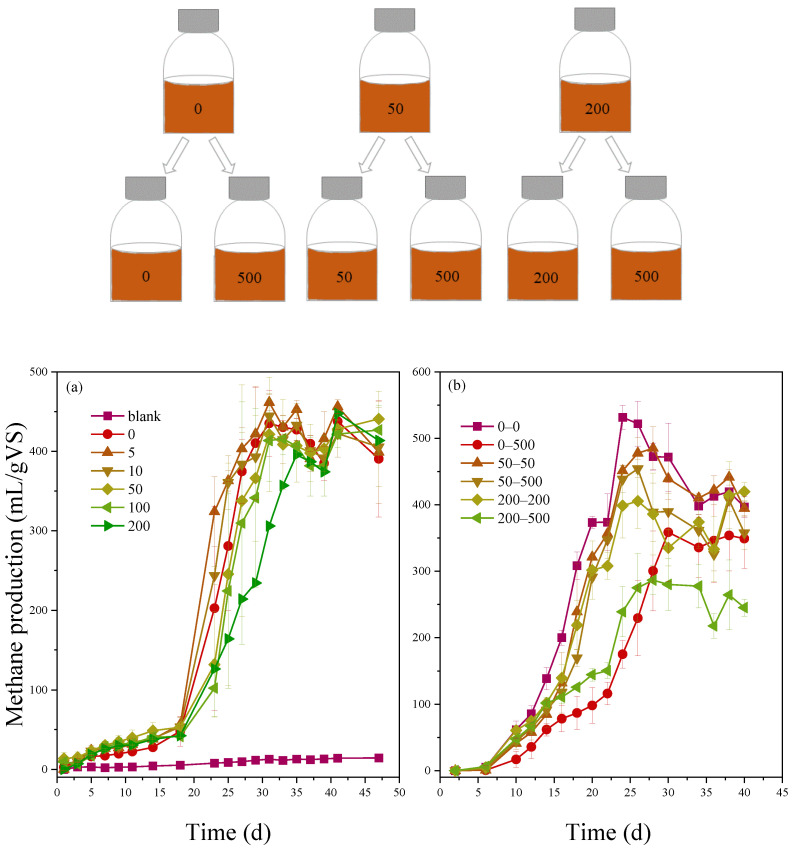
Effects of ERY on methane production during anaerobic digestion. (**a**) Phase I. (**b**) Phase II.

**Figure 2 molecules-29-03489-f002:**
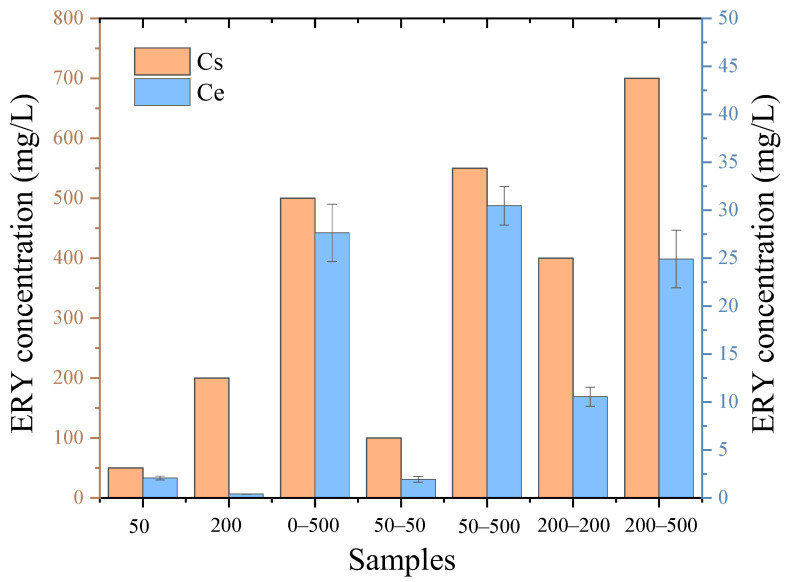
Variation in ERY concentration during anaerobic digestion (Cs and Ce represent initial and final concentrations of ERY, respectively).

**Figure 3 molecules-29-03489-f003:**
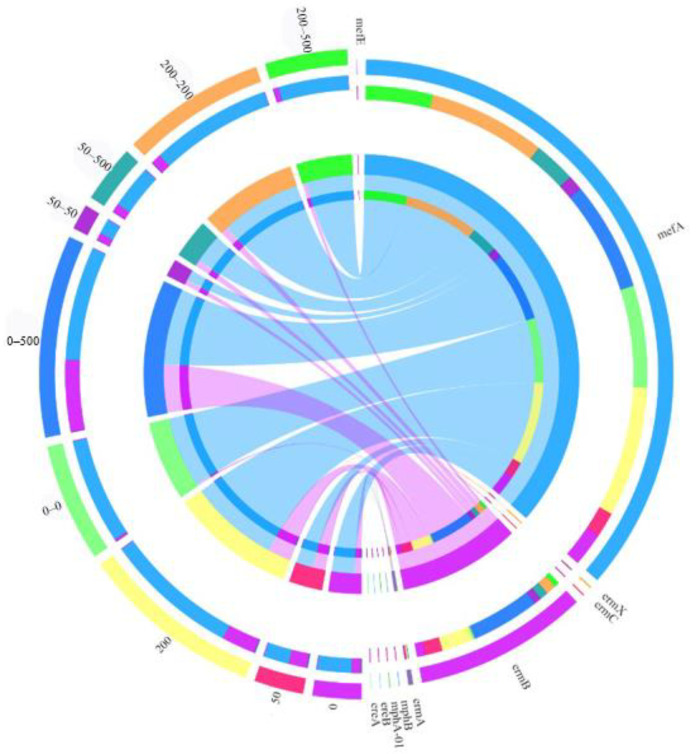
Chord diagram showing relative abundance of ARGs in different anaerobic digestion treatments.

**Figure 4 molecules-29-03489-f004:**
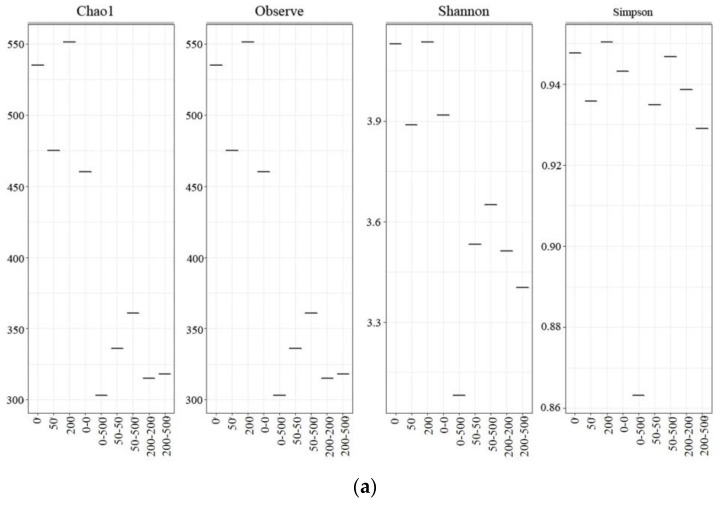
Variation in microbial community diversity and structure. (**a**) Alpha diversity. (**b**) Principal coordinate analysis (PCoA) based on Bray–Curtis distance showing distribution of microbial community.

**Figure 5 molecules-29-03489-f005:**
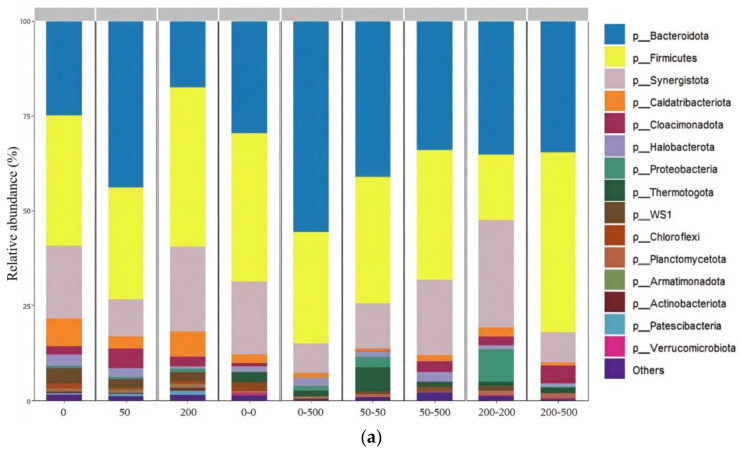
Changes in microbial community composition in different anaerobic digestion treatments. (**a**) Microbial community composition at phylum level. (**b**) Heatmap showing 50 genera with highest relative abundance in different anaerobic digestion treatments.

## Data Availability

The data are contained within the article or the Appendix A.

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
