# Peer review of "Effect of Stepwise Exposure to High-Level Erythromycin on Anaerobic Digestion"

_molecules, 2024, doi:10.3390/molecules29153489_

Round 1

Reviewer 1 Report

Comments and Suggestions for Authors

The authors aimed to evaluate the erythromycin removal efficiency of anaerobic digestion, exploring the influence of the erythromycin concentration on the methane production, microbial community structure and antibiotic resistance genes prevalence. Regardless of the ERY concentration anaerobic digestion showed removal efficiencies above 94%. After adaptation to low concentrations of ERY the anaerobic digestion process can maintain methane production levels. The presence of antibiotic stimulated the prevalence of ARGs genes, mainly ermB and mefA, although re-exposure to ERY promoted a decrease. The microbial community changes during the anaerobic digestion mirrored the occurring processes.

Antibiotic resistance is a relevant topic of worldwide concern and the efficient removal of antibiotics from wastewater is still changeling. Thus, the overall subject of the manuscript has scientific relevance, and fits the Molecules topics. The authors showed a good command of the English language. Minor editing of English language required.

Please see some suggestions for improvement below:

1.       The title is too long. Consider shortening it. Line 2: Change “of” to “to”. Line 4: add a “,” after “genes”.

2.       Line 16: Change “communities” to “community”.

3.       Line 21: Consider to change “whatever” to “regardless” for better flow.

4.       Line 23: Change “dominated” to “dominant”.

5.       Line 35: Although correct, consider change “annum” to “year”.

6.       Line 38: Change “study” to “studies”.

7.       Line 44: Change to “ …to remove ERY from wastewater..”

8.       In the Introduction section add information about:

a.       ERY residue worldwide,

b.       The importance of the ERY residue considering the One Health approach

c.       Information about the efficiency of removal of other processes

d.       The relevance of methane production

e.       Explain clearly the relevance of the study in the “real word” and not only on the laboratory scale.

9.       Figures captions must be self-explanatory. Please add information.

a.     Figure 1. The top scheme is not referred in the caption. Consider adjusting the Y-axis maximum to improve visibility of lower concentrations.

b.    Figue 2. Why did you use different graph types for the series? Clearly identify which axis refers to each series. No error bars/deviations?

c.     Figure 3. Add information to help the interpretation. “Abundances” or “Relative abundances”?

d.    Figure 4. The letters/numbers are too small. No information about the “G3” or the colors meaning. The PCA symbols are very small.

10.   Be careful with the extrapolation of the data, the error bars seem to overlapped.

11.   Did you performed any statistical analysis to evaluate the differences between conditions? If Yes give the values, if No discuss the limitations.

12.   Line 182: Change “…far away from 0” wording, for instance “Deviating considerably from zero”.

13.   Line 185: “insignificant” What do you mean? Have you performed statistical analysis?

14.   Line 193 – 196: Enumerate only de most dominant phyla. Remove “and so on”.

15.   Explain why Bacteroidota increased in the 50 mg/L but decreased in 200 mg/L.

16.   Have you performed any analysis to understand the significance/quantify of the substrate change?

17.   Please avoid the use of adjectives for quantification, such as “quite” (Line 260) that are subjective.

18.   Line 280: Change to “C37H67NO13”.

19.   Why did you changed the substrate in phase II.

20.   Subsection ARGs analysis

a.       RT-qPCR stands for Quantitative Reverse Transcription Polymerase Chain Reaction but you only extract DNA. Please clarify.

b.       What was the efficiency of reaction.

c.       What abundance normalization was performed.

d.       Give details on the procedure or at least the reference followed.

21.   Supplementary Materials:

a.       Gene numbers should be italicized

b.       Primer references are missing

22.    Authors should discuss the limitations of the study and the application to real scenarios.

23.   Line 329: Change “dominated” to “dominant”.

24.   Conclusions and abstract are very similar in writing please rewrite.

Comments on the Quality of English Language

The authors showed a good command of the English language. Minor editing of English language required.

Author Response

Comment 1: The title is too long. Consider shortening it. Line 2: Change “of” to “to”. Line 4: add a “,” after “genes”.

Response: We have revised the title.

Effect of stepwise exposure to high-level erythromycin on anaerobic digestion

Comment 2: Line 16: Change “communities” to “community”.

Response: We have changed “communities” into “community”.

Comment 3: Line 21: Consider to change “whatever” to “regardless” for better flow.

Response: We have changed “whatever” into “regardless of”.

Comment 4: Line 23: Change “dominated” to “dominant”.

Response: We have changed “dominated” into “dominant”.

Comment 5: Line 35: Although correct, consider change “annum” to “year”.

Response: We have changed “annum” into “year”.

Comment 6: Line 38: Change “study” to “studies”.

Response: We have changed “study” into “studies”.

Comment 7: Line 44: Change to “ …to remove ERY from wastewater..”

Response: We have changed “to remove ERY” into “to remove ERY from wastewater”.

Comment 8: In the Introduction section add information about:

  1. ERY residue worldwide,
  2. The importance of the ERY residue considering the One Health approach
  3. Information about the efficiency of removal of other processes
  4. The relevance of methane production
  5. Explain clearly the relevance of the study in the “real word” and not only on the laboratory scale.

Response: We have rewritten the introduction.

Erythromycin (ERY), a broad-spectrum macrolide antibiotic, is produced by fermentation of Streptomyces species [1]. Since ERY is frequently used in human medicine and veterinary medicine, its demand is high. For example, the usage of ERY in UK is high at 48 tonnes/year [2]. However, due to incomplete extraction during the ERY production process, its residue in the fermentation wastewater is high, reaching the 100–500 mg/L level [3]. ERY has been detected globally in sewage (max. conc. of 1037 μg/kg dry wt.), surface water (max. conc. of 3847 ng/L) and sediment (max. conc. of 175.38 μg/kg dry wt.) [4]. The accumulation of ERY in the environment may pose a significant risk to the ecosystem. Previous studies found that ERY could cause toxic effects on algae, such as Microcystis flosaquae [5] and Pseudokirchneriella subcapitata [6], as well as fish, such as Sparus aurata L. [7] and Oncorhynchus mykiss [8]. ERY is included on the US EPA’s Drinking Water Contaminant Candidate List 4 (CCL-4) [9] and the EU’s Water List [10]. Therefore, it is urgent to deal with the high-level ERY fermentation wastewater.

Various treatment methods have been used to remove ERY from wastewater [11-13]. For example, Gholamiyan et al. used magnetic-activated carbon as adsorbent for the removal of ERY and found that 95% of ERY could be adsorbed [11]. Mohammed et al. synthesized CuFe2O4@Chitosan for photocatalytic degradation of ERY and found high ERY degradation efficiency (90%) [12]. Chu et al. investigated the removal of ERY by ionizing radiation and found that ERY removal rate could reach 87% [13]. However, the above treatment methods are expensive. Besides, the high chemical oxygen demand (COD) and total suspended solids (TSS) in ERY fermentation wastewater limited their application in ERY wastewater treatment. Anaerobic digestion (AD) is considered as an economically feasible method to deal with recalcitrant pollutants owing to the low cost and high energy efficiency [14, 15]. Spielmeyer et al. [16] investigated the removal of tetracyclines and sulfonamides by anaerobic fermentation and found that their elimination rates ranged from 14% to 89%. It has been reported that the AD process could achieve 100% elimination of tetracycline [17].

Comment 9: Figures captions must be self-explanatory. Please add information.

  1. Figure 1. The top scheme is not referred in the caption. Consider adjusting the Y-axis maximum to improve visibility of lower concentrations.
  2. Figue 2. Why did you use different graph types for the series? Clearly identify which axis refers to each series. No error bars/deviations?
  3. Figure 3. Add information to help the interpretation. “Abundances” or “Relative abundances”?
  4. Figure 4. The letters/numbers are too small. No information about the “G3” or the colors meaning. The PCA symbols are very small.

Response: We have revised Figures.

Figure 1. Effects of ERY on methane production during anaerobic digestion. (a) Phase Ⅰ. (b) Phase Ⅱ.

Figure 2. Variation of ERY concentration during anaerobic digestion (Cs and Ce represented the initial and final concentrations of ERY, respectively).

The initial concentration of ERY was a fixed value and there is no error bar.

Figure 3. Chord diagram showing the relative abundances of ARGs in different anaerobic digestion treatments.

(a)

(b)

Figure 4. Variation of microbial community diversity and structure. (a) Alpha diversity. (b) Principal coordinate analysis (PCoA) based on the Bray-Curtis distance showing the distribution of microbial community.

Comment 10: Be careful with the extrapolation of the data, the error bars seem to overlapped.

Response: Thank you for your valuable suggestion. However, the data were real.

Comment 11: Did you performed any statistical analysis to evaluate the differences between conditions? If Yes give the values, if No discuss the limitations.

Response: Thank you for your valuable suggestion. During the experiment, we took several samples and combined them into a single sample for analysis. Thus, we did not perform any statistical analysis to evaluate the differences between conditions. In the future experiment, we will take several samples according to your suggestion and perform statistical analysis.

Comment 12: Line 182: Change “…far away from 0” wording, for instance “Deviating considerably from zero”.

Response: We have changed “…far away from 0” into “deviating considerably from 0”.

Comment 13: Line 185: “insignificant” What do you mean? Have you performed statistical analysis?

Response: We did not perform statistical analysis. We have revised the sentence.

Besides, PCoA did not reveal difference between 0-0 and 50-500, but revealed difference between 0-0 and 0-500.

Comment 14: Line 193 – 196: Enumerate only de most dominant phyla. Remove “and so on”.

Response: We have removed “and so on”.

Comment 15: Explain why Bacteroidota increased in the 50 mg/L but decreased in 200 mg/L.

Response: As shown in Fig. 5 (a), Bacteroidota increased in 50 but decreased in 200, which might be due to the fact that hydrolytic process associated with Bacteroidota was promoted at low-level ERY but inhibited at high-level ERY.

Comment 16: Have you performed any analysis to understand the significance/quantify of the substrate change?

Response: We did not perform any analysis to understand the significance/quantify of the substrate change. Actually, glucose is a monosaccharide, and avicel is a disaccharide. There is little difference in the effect of these two substrates on anaerobic fermentation. The difference between 0 and 0-0 was caused by the microbial acclimation. We have revised the manuscript.

Comment 17: Please avoid the use of adjectives for quantification, such as “quite” (Line 260) that are subjective.

Response: We have removed “quite”.

Comment 18: Line 280: Change to “C37H67NO13”.

Response: We have changed “C37H67NO13” into “C37H67NO13”.

Comment 19: Why did you changed the substrate in phase II.

Response: Actually, glucose is a monosaccharide, and avicel is a disaccharide. There is little difference in the effect of these two substrates on anaerobic fermentation. Compared with glucose, avicel has more advantages in alleviating acid inhibition caused by hydrolytic acidification.

Comment 20: Subsection ARGs analysis

  1. RT-qPCR stands for Quantitative Reverse Transcription Polymerase Chain Reaction but you only extract DNA. Please clarify.
  2. What was the efficiency of reaction.
  3. What abundance normalization was performed.
  4. Give details on the procedure or at least the reference followed.

Response: We have written the ARGs analysis.

The detailed composition of PCR mixture and PCR procedures were processed according to a previous study [52]. The efficiency of amplification should be within 1.8-2.2. The relative abundance of ARG was normalized to 16S rRNA gene, which was measured according to a comparative CT method, 2−ΔCT where ΔCT = (CTARG – CT16S rRNA gene).

[52] Zhang, Y., Liu, H., Dai, X., Cai, C., Wang, J., Wang, M., Shen, Y., Wang, P. 2020. Impact of application of heat-activated persulfate oxidation treated erythromycin fermentation residue as a soil amendment: Soil chemical properties and antibiotic resistance. Sci. Total Environ., 736, 139668.

Comment 21: Supplementary Materials:

  1. Gene numbers should be italicized
  2. Primer references are missing

Response: We have revised the Supplementary Materials.

Table S1. qPCR primer sequences for antibiotic resistance genes analysis.

Resistance mechanism

gene

5' to 3'

Reference

esterase gene

ereA

F: CCTGTGGTACGGAGAATTCATGT

[1-3]

R: ACCGCATTCGCTTTGCTT

ereB

F: GCTTTATTTCAGGAGGCGGAAT

R: TTTTAAATGCCACAGCACAGAATC

phosphorylase gene

mphA-01

F: CTGACGCGCTCCGTGTT

R: GGTGGTGCATGGCGATCT

mphB

F: CGCAGCGCTTGATCTTGTAG

R: TTACTGCATCCATACGCTGCTT

methylase gene

ermA

F: TTGAGAAGGGATTTGCGAAAAG

R: ATATCCATCTCCACCATTAATAGTAAACC

ermB

F: TAAAGGGCATTTAACGACGAAA

R: TTTATACCTCTGTTTGTTAGGGAATTGAA

ermC

F: TTTGAAATCGGCTCAGGAAAA

R: ATGGTCTATTTCAATGGCAGTTACG

ermX

F: GCTCAGTGGTCCCCATGGT

R: ATCCCCCCGTCAACGTTT

efflux pump gene

mefA

F: CCGTAGCATTGGAACAGCTTTT

R: AAACGGAGTATAAGAGTGCTGCAA

mefE

F: CGTATTGGGTGCTGTGATTG

R: TATGCACAGGCGTTCCATTA

References

[1] M.T. Guo, Q.B. Yuan, J. Yang, Ultraviolet reduction of erythromycin and tetracycline resistant heterotrophic bacteria and their resistance genes in municipal wastewater, Chemosphere, 93 (2013) 2864-2868.

[2] M. Liu, R. Ding, Y. Zhang, Y. Gao, Z. Tian, T. Zhang, M. Yang, Abundance and distribution of Macrolide-Lincosamide-Streptogramin resistance genes in an anaerobic-aerobic system treating spiramycin production wastewater, Water Res, 63 (2014) 33-41.

[3] Y. Chen, P. Li, Y. Huang, K. Yu, H. Chen, K. Cui, Q. Huang, J. Zhang, K. Yew-Hoong Gin, Y. He, Environmental media exert a bottleneck in driving the dynamics of antibiotic resistance genes in modern aquatic environment, Water Res, 162 (2019) 127-138.

Comment 22: Authors should discuss the limitations of the study and the application to real scenarios.

Response: There are still some limitations in this study: the role of transformation products of ERY on the fate of ARGs and microorganisms need further research, and the application of AD for the treatment of the actual antibiotic fermentation wastewater should be clarified.

Comment 23: Line 329: Change “dominated” to “dominant”.

Response: We have changed “dominated” into “dominant”.

Comment 24: Conclusions and abstract are very similar in writing please rewrite.

Response: We have rewritten the conclusions.

This study investigated the influence of stepwise exposure of high-level ERY on anaerobic digestion, and the main conclusions were as follows: (1) After domestication with low-level ERY, AD could adapt to the presence of high-level ERY and could maintain efficient CH4 production. (2) The AD process could achieve higher removal of ERY. (3) The first exposure of ERY stimulated the increase of total ARGs abundance, while the AD process seemed to discourage ARGs maintenance following re-exposure to ERY. (4) Under the influence of ERY, the hydrogenotrophic methanogenesis pathway replaced the acetoclastic methanogenesis pathway for CH4 production. There are still some limitations in this study: the role of transformation products of ERY on the fate of ARGs and microorganisms need further research, and the application of AD for the treatment of the actual antibiotic fermentation wastewater should be clarified.

Reviewer 2 Report

Comments and Suggestions for Authors

The authors researched the removal of erythromycins from wastewater using anaerobic digestion. The work is in line with current trends in searching for effective methods of purifying pharmaceutical compounds. The work is interesting for readers and presents promising results. The reviewer indicates his comments below, which will contribute to improving the quality of the article:

·        Lines 36-37 „its residue in the fermentation wastewater is really high, reaching the 100–500

·        mg/L level [3]” -remove the word really.

·        Introduction - Add information about ERY concentrations in sewage, surface water, sediment and living organisms. This way you will show the scale of the threat to the aquatic ecosystem, as well as the threat to human health.

·        Lines 43-44 „Various treatment methods, including ionizing, photocatalysis, and adsorption have

·        been used to remove ERY [10-12].” - expand on the topic of conventional methods of treating wastewater rich in ERY and advanced oxidation methods. Indicate what percentages scientists achieved in reducing ERY and what are the disadvantages of the methods used. In this way, the reader will be better introduced to why the authors focused on anaerobic treatment.

·        2.2. The removal of ERY during AD - transformation products may be formed during the ERA degradation process. Have the authors attempted to identify transformation products? This may be interesting because the purification process may produce products that will interact with microorganisms in a different way than ERY. If the authors did not study the transformation products, they may mention the above information as a hypothesis and propose in a future research plan that such an analysis may be helpful.

·        Figure 2. - the description of the vertical axes is the same "ERY concentration [mg/L]", however, the numerical values ​​are different. Please correct the axis description.

·        Figure 4- Can authors enlarge the legend and tags in the figure?

·        3.4. Analytical methods - please write what column and temperature program was used to determine methane using the GC method. Please describe how the calibration curve was prepared and provide the basic validation parameters of the method. Additionally, please provide the column, which solvents formed the mobile phase, what the elution program was, and what the flow rate was in the HPLC method. Please provide how the ERY was identified and how the calibration curve was prepared, as well as the basic validation parameters.

·        4. Conclusions- what are the future research plans related to the research topic described? Do the authors intend to conduct an experiment on real sewage or on a larger scale?

Comments on the Quality of English Language

Minor editing of English language required.

Author Response

Comment 1: Lines 36-37 „its residue in the fermentation wastewater is really high, reaching the 100–500 mg/L level [3]” -remove the word really.

Response: We have removed the word.

Comment 2: Introduction - Add information about ERY concentrations in sewage, surface water, sediment and living organisms. This way you will show the scale of the threat to the aquatic ecosystem, as well as the threat to human health.

Response: We have rewritten the introduction.

Erythromycin (ERY), a broad-spectrum macrolide antibiotic, is produced by fermentation of Streptomyces species [1]. Since ERY is frequently used in human medicine and veterinary medicine, its demand is high. For example, the usage of ERY in UK is high at 48 tonnes/year [2]. However, due to incomplete extraction during the ERY production process, its residue in the fermentation wastewater is high, reaching the 100–500 mg/L level [3]. ERY has been detected globally in sewage (max. conc. of 1037 μg/kg dry wt.), surface water (max. conc. of 3847 ng/L) and sediment (max. conc. of 175.38 μg/kg dry wt.) [4]. The accumulation of ERY in the environment may pose a significant risk to the ecosystem. Previous studies found that ERY could cause toxic effects on algae, such as Microcystis flosaquae [5] and Pseudokirchneriella subcapitata [6], as well as fish, such as Sparus aurata L. [7] and Oncorhynchus mykiss [8]. ERY is included on the US EPA’s Drinking Water Contaminant Candidate List 4 (CCL-4) [9] and the EU’s Water List [10]. Therefore, it is urgent to deal with the high-level ERY fermentation wastewater.

Comment 3: Lines 43-44 „Various treatment methods, including ionizing, photocatalysis, and adsorption have been used to remove ERY [10-12].” - expand on the topic of conventional methods of treating wastewater rich in ERY and advanced oxidation methods. Indicate what percentages scientists achieved in reducing ERY and what are the disadvantages of the methods used. In this way, the reader will be better introduced to why the authors focused on anaerobic treatment.

Response: We have rewritten the introduction.

Various treatment methods have been used to remove ERY from wastewater [11-13]. For example, Gholamiyan et al. used magnetic-activated carbon as adsorbent for the removal of ERY and found that 95% of ERY could be adsorbed [11]. Mohammed et al. synthesized CuFe2O4@Chitosan for photocatalytic degradation of ERY and found high ERY degradation efficiency (90%) [12]. Chu et al. investigated the removal of ERY by ionizing radiation and found that ERY removal rate could reach 87% [13]. However, the above treatment methods are expensive. Besides, the high chemical oxygen demand (COD) and total suspended solids (TSS) in ERY fermentation wastewater limited their application in ERY wastewater treatment. Anaerobic digestion (AD) is considered as an economically feasible method to deal with recalcitrant pollutants owing to the low cost and high energy efficiency [14, 15]. Spielmeyer et al. [16] investigated the removal of tetracyclines and sulfonamides by anaerobic fermentation and found that their elimination rates ranged from 14% to 89%. It has been reported that the AD process could achieve 100% elimination of tetracycline [17].

Comment 4: 2.2. The removal of ERY during AD - transformation products may be formed during the ERA degradation process. Have the authors attempted to identify transformation products? This may be interesting because the purification process may produce products that will interact with microorganisms in a different way than ERY. If the authors did not study the transformation products, they may mention the above information as a hypothesis and propose in a future research plan that such an analysis may be helpful.

Response: Thank you for your valuable suggestion. We did not study the transformation products. However, we agree with your opinion that this is an interesting topic. We will study the transformation products in future studies.

Comment 5: Figure 2. - the description of the vertical axes is the same "ERY concentration [mg/L]", however, the numerical values are different. Please correct the axis description.

Response: We have revised Figure 2.

Figure 2. Variation of ERY concentration during anaerobic digestion (Cs and Ce represented the initial and final concentrations of ERY, respectively).

Comment 6: Figure 4- Can authors enlarge the legend and tags in the figure?

Response: We have revised Figure 4.

(a)

(b)

Figure 4. Variation of microbial community diversity and structure. (a) Alpha diversity. (b) Principal coordinate analysis (PCoA) based on the Bray-Curtis distance showing the distribution of microbial community.

Comment 7: 3.4. Analytical methods - please write what column and temperature program was used to determine methane using the GC method. Please describe how the calibration curve was prepared and provide the basic validation parameters of the method. Additionally, please provide the column, which solvents formed the mobile phase, what the elution program was, and what the flow rate was in the HPLC method. Please provide how the ERY was identified and how the calibration curve was prepared, as well as the basic validation parameters.

Response: We have written the analytical methods.

TS and VS were analyzed according to [53]. The pH was measured by a pH meter (Radio-meterTM). The content of CH4 was quantified by a gas-chromatograph (Trace 1310 GC-TCD, Thermo Fisher, Denmark) equipped with a thermal conductivity detector (TCD) and Thermo (P/N 26004–6030) Column (30 m length, 0.320 mm inner diameter, and film thickness 10 µm) with helium as carrier gas. The concentration of ERY was determined using a high-performance liquid chromatography (Agilent 1290 Infinity, USA, HPLC) system coupled with a tandem mass spectrometer (Agilent 6470 series, USA, MS/MS). The system consisted of electrospray ionization (ESI) and C18 column (2.1 × 50 mm, 1.7 μm). The positive ion mode (ESI+) was selected. ERY was detected by the prominent fragment ion at m/z 158. Mobile phase A was MeOH and mobile phase B was water + 0.1% formic acid (at the flow rate of 0.2 mL/min for 8 min). The elution gradient mode was as follows: 45% A (0-2.5 min), 25% A (2.5-4 min), 10% A (4-7 min), and 25% A (7-8 min). Principal coordinate analysis (PCoA), heatmap and chord diagram were conducted using R soft-ware. OriginPro 2021 was used to analyze other figures.

Comment 8: 4. Conclusions- what are the future research plans related to the research topic described? Do the authors intend to conduct an experiment on real sewage or on a larger scale?

Response: We have rewritten the conclusions.

This study investigated the influence of stepwise exposure of high-level ERY on anaerobic digestion, and the main conclusions were as follows: (1) After domestication with low-level ERY, AD could adapt to the presence of high-level ERY and could maintain efficient CH4 production. (2) The AD process could achieve higher removal of ERY. (3) The first exposure of ERY stimulated the increase of total ARGs abundance, while the AD process seemed to discourage ARGs maintenance following re-exposure to ERY. (4) Under the influence of ERY, the hydrogenotrophic methanogenesis pathway replaced the acetoclastic methanogenesis pathway for CH4 production. There are still some limitations in this study: the role of transformation products of ERY on the fate of ARGs and microorganisms need further research, and the application of AD for the treatment of the actual antibiotic fermentation wastewater should be clarified.

Round 2

Reviewer 2 Report

Comments and Suggestions for Authors

The authors responded to all the reviewer's comments and doubts. The authors made every effort and put a lot of effort into creating the final version of the manuscript. According to the reviewer, the article in its current form is of adequate quality and can be published.